# Boosting Fast and High-Quality Speech Synthesis with Linear Diffusion

## Abstract

Denoising Diffusion Probabilistic Models have shown extraordinary ability on various generative tasks. However, their slow inference speed renders them impractical in speech synthesis. This paper proposes a linear diffusion model (LinDiff) based on an ordinary differential equation to simultaneously reach fast inference and high sample quality. Firstly, we employ linear interpolation between the target and noise to design a diffusion sequence for training, while previously the diffusion path that links the noise and target is a curved segment. When decreasing the number of sampling steps (i.e., the number of line segments used to fit the path), the ease of fitting straight lines compared to curves allows us to generate higher quality samples from a random noise with fewer iterations. Secondly, To reduce computational complexity and achieve effective global modeling of noisy speech, LinDiff employs a patch-based processing approach that partitions the input signal into small patches. The patch-wise token leverages Transformer architecture for effective modeling of global information. Adversarial training is used to further improve the sample quality with decreased sampling steps. We test proposed method with speech synthesis conditioned on acoustic feature (Mel-spectrograms). Experimental results verify that our model can synthesize high-quality speech even with only one diffusion step. Both subjective and objective evaluations demonstrate that our model can synthesize speech of a quality comparable to that of autoregressive models with faster synthesis speed (3 diffusion steps).

## 1 Introduction

Deep generative models have made tremendous strides in the realm of speech synthesis. Overall, contemporary speech synthesis techniques can be broadly categorized into two paradigms: methods that leverage likelihood-based modeling and methods based on generative adversarial networks (GANs). For example, WaveNet (Oord et al., 2016), an autoregressive likelihood-based model, can synthesize high-quality speech. However, it is also characterized by expensive computational cost at inference time. Moreover, alternative approaches such as Flow-based model (Prenger et al., 2019) and Variational AutoEncoders (VAE) (Kingma et al., 2019) have their own limitations on sample quality. While GAN-based models (Goodfellow et al., 2020; Kumar et al., 2019; Kong et al., 2020a) exhibit fast-paced speech synthesis, they are concurrently beset by training instability and limited sample diversity.

An emerging group of generative models, Denoising Diffusion Probabilistic Models (DDPMs) (Ho et al., 2020; Song et al., 2020), a likelihood-based model, have become increasingly popular in speech synthesis. For instance, WaveGrad (Chen et al., 2020) and DiffWave (Kong et al., 2020b) produce high-quality samples that match the quality of autoregressive methods. However, the iterative optimization in DDPMs significantly slows down sampling speed. To tackle this problem, existing approaches either design an extra structure such as a noise schedule network (Lam et al., 2022; Huang et al., 2022) or describe the random diffusion process with the ordinary differential equation (ODE) (Liu et al., 2022). However, ODE-based diffusion still needs several steps to produce high-fidelity sample.

Inspired by rectified flow (Liu et al., 2022), We proposed a conditional diffusion model. During the training process, we performs a linear interpolation between a target sample and initial standard

Gaussian noise to construct the diffusion sequence and train the denoising network to fit the path reversely. For the inference process, we reconstruct the target through the Euler sampling method from a randomly sampled standard Gaussian noise.

In the light of the success of Vision Transformer (ViT) (Bao et al., 2022) for image synthesis, we propose a similar structure for audio that turns continual sampling points into an audio patch and apply Transformer (Vaswani et al., 2017) to build contextual connections for these tokens. We then use a Time-Aware Location-Variable Convolution (Huang et al., 2022) module for fine-grained detail restoration. As demonstrated in previous work (Xiao et al., 2022), the combination of DDPMs and GANs has shown promising performance. In line with this, we incorporate adversarial training into our method to enhance the quality of generated samples while reducing the number of required iteration steps.

Overall, the main contributions of this paper are:

- A linear conditional diffusion algorithm with an ordinary differential equation is proposed to reduce the steps required during inference. Experiments demonstrate this method can synthesis relatively high-fidelity speech with limited steps.
- A Transformer-based architecture for audio denoising is introduced. As Transformer captures long in-context information efficiently, it enables rapid enlargement of the receptive field. To the best of our knowledge, we are the first to apply Transformer for conditional waveform generation (i.e., vocoder).
- Adopting implicit diffusion to combine Linear diffusion with adversarial training for further reducing the steps for inference while maintaining the generated speech's high quality. Experiments show the introduction of adversarial training enables the proposed model to synthesize relatively high-fidelity speech even with only 1 step.

## 2 BACKGROUND

Our proposed method is based on Flow-Matching (Lipman et al., 2023) and Rectified-Flow (Liu et al., 2022). We first introduce Continuous Normalizing Flow (CNF) (Chen et al., 2019). Using $x \in \mathbb{R}^d$ to represent data points in data space. Defining $p : [0,1] \times \mathbb{R}^d \to \mathbb{R}^d$ the time-dependent probability density path. Defining $v : [0,1] \times \mathbb{R}^d \to \mathbb{R}^d$ the time-dependent vector field. Defining $\phi : [0,1] \times \mathbb{R}^d \to \mathbb{R}^d$ the time-dependent diffeomorphic map. These variables satisfy,

$$\frac{d\phi_t(x)}{dt} = v_t(\phi_t(x)). \tag{1}$$

A CNF is used to reshape a simple prior density $p_0$ to a more complicated one via the push-forward equation,

$$p_t = [\phi_t] * p_0, \tag{2}$$

where $*$ is defined by,

$$[\phi_t] * p_0(x) = p_0(\phi^{-1}(x)) det[\frac{\partial \phi_t^{-1}}{\partial x}(x)]. \tag{3}$$

Let $x_1$ denote a random variable from unknown data distribution $q(x_1)$. It is assumed we only have access to data samples from $q(x_1)$ but not the density function itself. Here we let $p_t$ be a probability path such that $p_0 = p$ is a simple distribution (standard normal distribution) and let $p_1$ be approximately equal in distribution to $q$. Given target probability density path $p_t(x)$ and a corresponding vector field $u_t(x)$ , which generates $p_t(x)$, (Lipman et al., 2023) define the Flow Matching (FM) objective as,

$$\mathcal{L}_{FM}(\theta) = \mathbb{E}_{t,p_t(x)}||v_t(x) - u_t(x)||^2, \tag{4}$$

where $\theta$ denotes the learnable parameters of the CNF vector field $v_t$ (as defined before), $t \sim U[0,1]$ and $x \sim p_t(x)$. Though FM objective is simple and attractive, it is intractable to use in practice on its own since no prior knowledge for what an appropriate $p_t$ and $u_t$ are. (Lipman et al., 2023) build $p_t, u_t$ from conditional probability paths and vector fields. Specially, given a particular data sample $x_1$, using $p_t(x|x_1)$ to denote a conditional probability path so that it satisfies $p_0(x|x_1) = p(x)$ at

$t = 0$. The authors design $p_1(x|x_1) = N(x|x_1, \sigma^2 I)$, a normal distribution with $X_1$ mean and a sufficiently small standard deviation. Then we get following equations,

$$p_t(x) = \int p_t(x|x_1)q(x_1)dx_1, \tag{5}$$

where at time $t = 1$, as $p_1(x|x_1)$ is defined to be a distribution concentrated around $x_1$, we can take this function as the unit impulse response function. Then the marginal probability $p_1$ is a mixture distribution that closely approximates the data distribution $q$,

$$p_t(x) = \int p_t(x|x_1)q(x_1)dx_1 \approx q(x). \tag{6}$$

The authors also define a conditional vector field that generates $p_t(\cdot|x_1)$,

$$u_t(x) = \int u_t(x|x_1)\frac{p_t(x|x_1)q(x_1)}{p_t(x)}dx_1, \tag{7}$$

(Lipman et al., 2023) proves that $u_t(x)$ defined before generates the marginal probability path $p_t()x$. They also define the Conditional Flow Matching objective (CFM),

$$\mathcal{L}_{CFM}(\theta) = \mathbb{E}_{t,q(x_1),p_t(x|x_1)}||v_t(x) - u_t(x|x_1)||^2, \tag{8}$$

where $t \sim U[0,1], x_1 \sim q(x_1)$ and now $x \sim p_t(x|x_1)$. FM and CFM were proved have identical gradients. In other words, optimizing the CFM objective is equivalent to optimizing the FM objective. (Lipman et al., 2023) compares this method with Score Matching (DDPM), finding it has constant direction in time and is arguably simpler to fit with a parametric model. This means decreasing the sampling steps in this method suffers less than in DDPM. In our method, we build the probability path with each sample $x_1$ with a stand normal distribution $N(x|0, I)$ and optimize the CFM objective.

## 3 METHOD

In this section, we present a comprehensive overview of our proposed methodology.

### 3.1 LINEAR DIFFUSION

Assumping the data $x_1$ means target (ground truth audio or image), We use $x_0$ to denote the original noise sampled from $p_0(x|x_1) \sim N(x|0, I)$. We construct a probability path from $x_0$ to $x_1$ through linear interpolation. In other words,

$$x_t = t\frac{x_1 - x_0}{T} + x_0, \tag{9}$$

where $T = 1$ and $t \sim [0, 1]$. Also the probability path $p_t(x|x_1)$ is,

$$p_t(x|x_1) = N(x|tx_1, (1-t)^2 I), \tag{10}$$

According to eq 1 the conditional vector field is,

$$u_t(x|x_1) = \frac{x_1 - x_0}{T}, \tag{11}$$

Then $\mathcal{L}_{CFM}$ satisfies the following equation,

$$\mathcal{L}_{CFM}(\theta) = \mathbb{E}_{t,q(x_1),p_t(x|x_1)}||v_t(x) - \frac{x_1 - x_0}{T}||^2, \tag{12}$$

We interpolate 999 points between 0 and 1, thus building a probability path (diffusion path).

### 3.2 LINDIFF

LinDiff is a backbone that combines Transformer (Vaswani et al., 2017) and Convolutional Neural Network for diffusion-based speech synthesis. We also apply discriminators for the training process. Specifically, LinDiff parameterizes the noise prediction network $v_t(x)$ in Eq. 14. It takes the diffusion step $t$, the condition $\mathbf{c}$ and the noisy audio $\mathbf{a}_t^{rev}$ as inputs and predict the target speech $\mathbf{a}_T^{rev}$.

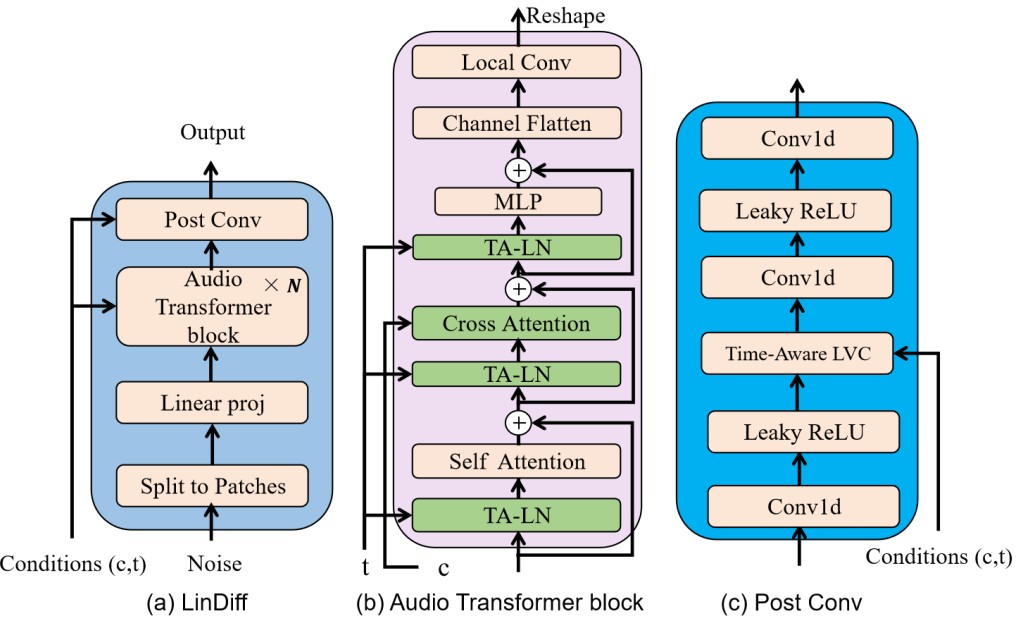

Figure 1: The overall architecture of LinDiff. TA-LN represents TimeAdaptive LayerNorm. Time-Aware LVC represents Time-Aware Location-Variable Convolution module proposed by Huang et al. (2022).

**Audio Transformer block**    Inspired by the U-ViT backbone in diffusion models (Bao et al., 2022), we introduce an Audio Transformer (AiT) block for speech synthesis. To achieve this, we partition the input noise into smaller patches, treating each patch as a token. Subsequently, we apply a linear transformation to obtain patch embeddings, which are then fed into the Audio Transformer block. In our experimentation, we explore various patch sizes and find that reducing the patch size improves model performance. However, it is important to note that this improvement comes at the expense of increased computational cost due to the higher number of patches. Thus, a trade-off between model performance and computational efficiency must be carefully considered when selecting the optimal patch size. In our implementation, we set the patch size to be 64.

**Feature fusion**    For each time step $t$, we follow paper (Zeng et al., 2021) to embed the step into an 128-dimensional positional encoding vector $\mathbf{e}_t$ and then apply linear transformation to turn it into diffusion-step embedding $\mathbf{t}_{emb}$. We propose Time-Adaptive Layer Norm to fuse the step information. Supposing the noise audio feature is $\mathbf{x}$ and LN denote layer normalization (Ba et al., 2016).

$$\text{TALN}(\mathbf{x}, \mathbf{t}_{emb}) = g(\mathbf{t}_{emb}) \cdot \text{LN}(\mathbf{x}) + b(\mathbf{t}_{emb}). \tag{13}$$

To fuse the accoustic feature, a cross attention module was employed. We first use linear transformation to turn it into hidden features. And then the input noisy audio feature sequence will assume the role of query, interacting with the Mel hidden feature sequence.

**Post Conv**    Due to partitioning the input sequence into small patches, the resolution of the model is limited and the output sequence obtained through AiT losses much high-frequency information. However, as the human ear is sensitive to speech, we here add a Post Convolution module to process the details of the output. We follow Huang et al. (2022) to use a Time-Aware Location-Variable Convolution module with some simple Conv1d layer as our Post Conv module. Experimental results demonstrate that this approach improves the quality of the audio.

### 3.3    TRAINING LOSS

The aim of this work is to propose a model with fast inference speed, high-quality speech generation capability and training stability. However, reducing the number of sampling steps in our models

led to a decrease in the quality of generated speech, similar to the findings in existing literature. To address this issue, we draw inspiration from the DiffGAN (Xiao et al., 2022) and introduce the adversarial training scheme into our model. This allows us to maintain high-quality speech while reducing the number of iterations. Additionally, the presence of the diffusion process improves the stability of adversarial training. For the existing work, (Xiao et al., 2022) parameterize the denoising function as an implicit denoising model. We follow this way. Specifically, instead of calculating $\mathbf{a}_{t+1}^{rev}$ directly from $\mathbf{a}_t^{rev}$, we first predict $\mathbf{a}_T^{rev}$ (The target waveform) and then obtain $\mathbf{a}_{t+1}^{rev}$ with following formulation (as we interpolate 999 points between origin noise and target):

$$\mathbf{a}_{t+1}^{rev} = \mathbf{a}_t^{rev} + (\mathbf{a}_T^{rev} - \mathbf{a}_0^{rev})\frac{1}{1000}. \tag{14}$$

Our discriminator is denoted as $D_\phi(\mathbf{a}_T^{rev})$, where $\mathbf{a}_T^{rev}$ denotes the predicted blurry audio. We apply three discriminators from different perspectives on $\mathbf{a}_T^{rev}$ predicted from each time step. One of the discriminators examines the samples from a spectral perspective, while the other two adopt the multi-scale and multi-period discriminators utilized in HiFiGAN (Kong et al., 2020a), both of them perform discrimination in the time domain.

Our loss consists of three parts: diffusion loss, frequency-domain reconstruction loss, and adversarial loss.

We use wav to represent the target. The Diffusion loss is,

$$\mathcal{L}_{diff} = \text{MSE}(\mathbf{wav}_{gt} - \mathbf{a}_T^{rev}), \tag{15}$$

This seems to be different from eq 12. As we directly predict the target, the vector field can be obtained indirectly through,

$$v_t(x) = \frac{\mathbf{a}_T^{rev} - \mathbf{a}_0^{rev}}{T}, \tag{16}$$

So Eq 15 and Eq 12 are actually equivalent.

We use STFT to represent Short-time Fourier Transform. Then Frequency-domain reconstruction loss:

$$\mathcal{L}_s = \text{MSE}(\text{STFT}(\mathbf{wav}_{gt}) - \text{STFT}(\mathbf{a}_T^{rev})), \tag{17}$$

We set four pairs of ($window\_size, hop\_length, n\_fft$) and randomly choose one of them to carry out the STFT to alleviate overfitting issue.

We assume $D_\phi(\mathbf{a}_T^{rev})$ to represent the discriminator. For the generator, adversarial loss:

$$\mathcal{L}_{adv}^g = (1 - D_\phi(\mathbf{a}_T^{rev}))^2. \tag{18}$$

We use $\text{sg}(\cdot)$ to represent stop gradient, then for the discriminator:

$$\mathcal{L}_{adv}^d = D_\phi^2(\text{sg}(\mathbf{a}_T^{rev})) + (1 - D_\phi(\mathbf{a}_T^{true}))^2. \tag{19}$$

Total loss for generator is:

$$\mathcal{L}_{gen} = \mathcal{L}_{adv}^g + \mathcal{L}_s + \mathcal{L}_{diff}. \tag{20}$$

It is worth mentioning that our discriminator is composed of three sub-discriminators, resulting in a relatively high computational cost that slows down the training process. In order to expedite the training, the discriminator is updated every 5 training steps. To maintain a balanced adversarial training process for the generator, a weight of 0.2 was added on its adversarial loss.

## 3.4 ALGORITHM

We present a concise summary of the pseudocode outlining the training and inference processes of our model. The training process 1 consists of three stages. In the first stage, we sample short audio clips and train the model without adversarial training. In the second stage, we introduce adversarial training while still using short audio clips, and we update the discriminator's weights at each step. In the third stage, we sample long audio clips for training. Regarding the inference process 2, we begin by randomly sampling noise from a Gaussian standard distribution. Assuming the Mel spectrogram's size is $T \times 80$, the sampled noise has a size of $(256 \times T) \times 1$, which corresponds to the result of the Short-Time Fourier Transform (STFT) applied during the generation of the training dataset.

---

**Algorithm 1** LinDiff Training Algorithm

---

1: **Input**: Speech and correlated Mel-spectrograms ($\mathbf{a}_T$, **condition**), Random Gaussian noise $\mathbf{a}_0^{rev}$, Total diffusion steps $T$
2: **repeat**
3:     Randomly select a step $t \in [0, T)$
4:     Calculate $\mathbf{a}_t^{rev}$ with sampled $t$, noise $\mathbf{a}_0^{rev}$ and ground truth wavform $\mathbf{a}_T$ according to Eq 9
5:     Predict the target $\mathbf{a}_T^{rev} = f(\mathbf{a}_t^{rev}, \textbf{condition}, t)$
6:     **if** Stage 2 **then**
7:         Calculate the discriminator's loss $\mathcal{L}_{adv}^d = D_\phi^2(\mathrm{sg}(\mathbf{a}_T^{rev})) + (1 - D_\phi(\mathbf{a}_T))^2$
8:         Perform backpropagation on $\mathcal{L}_{adv}^d$ and update the weights of the discriminator
9:         Calculate LinDiff's loss $\mathcal{L}_{gen} = (1 - D_\phi(\mathbf{a}_T^{rev}))^2 + \mathrm{MSE}(\mathbf{a}_T^{rev}, \mathbf{a}_T) + \mathrm{MSE}(\mathrm{STFT}(\mathbf{a}_T^{rev}), \mathrm{STFT}(\mathbf{a}_T))$
10:         Perform backpropagation on $\mathcal{L}_{gen}$ and update the weights of LinDiff
11:     **end if**
12:     **if** Stage 3 **then**
13:         **if** $Step \% 5 == 0$ **then**
14:             Calculate the discriminator's loss $\mathcal{L}_{adv}^d = D_\phi^2(\mathrm{sg}(\mathbf{a}_T^{rev})) + (1 - D_\phi(\mathbf{a}_T))^2$
15:             Perform backpropagation on $\mathcal{L}_{adv}^d$ and update the weights of the discriminator
16:         **end if**
17:         Calculate LinDiff's loss $\mathcal{L}_{gen} = (1 - D_\phi(\mathbf{a}_T^{rev}))^2 + \mathrm{MSE}(\mathbf{a}_T^{rev}, \mathbf{a}_T) + 0.2 \times \mathrm{MSE}(\mathrm{STFT}(\mathbf{a}_T^{rev}), \mathrm{STFT}(\mathbf{a}_T))$
18:         Perform backpropagation on $\mathcal{L}_{gen}$ and update the weights of LinDiff
19:     **end if**
20: **until** convergence

---

**Algorithm 2** LinDiff inference algorithm

---

1: **Input**: Random noise $\mathbf{a}_0^{rev}$, Mel-spectrogram (**condition**), Total diffusion steps $T$
2: Set $t = 0$
3: **for** $t < T$ **do**
4:     Predict blurry target $\mathbf{a}_T^{rev} = f(\mathbf{a}_t, \textbf{condition}, t)$
5:     Caculate $\mathbf{a}_{t+1}^{rev}$ with $\mathbf{a}_t^{rev}, \mathbf{a}_T^{rev}$ and $\mathbf{a}_0^{rev}$ according to Eq 14
6:     $t = t + 0.001$ (as we interpolate 999 points between origin noise and target)
7: **end for**
8: **return** $\mathbf{a}_T^{rev}$

---

## 4 EXPERIMENTS

### 4.1 SETUP

**Datasets** In this study, we evaluated the proposed model on two distinct datasets. The first dataset is the LJ Speech dataset (Ito & Johnson, 2017), which is composed of 13,100 audio clips at a sampling rate of 22050 Hz, spoken by a single speaker reading passages from 7 non-fiction books. This dataset spans approximately 24 hours of audio in total. The second dataset is the LibriTTS dataset (Zen et al., 2019), which contains 585 hours of speech data from 2484 speakers. In all of the experiments, we utilized a 16-bit, 22050 Hz sampling rate. For the speech synthesis task, we used 80-band Mel-spectrograms as the condition. These spectrograms were extracted using Hann windowing with a frame shift of 12.5-ms, frame length of 50-ms, and a 1024-point Fourier transform.

**Model Configurations** The LinDiff model comprises three key components: a patch-embedding module, an Audio Transformer, and a Post-Conv module. Specifically, we utilize a patch size of 64 and apply a linear transformation to the patch-embedding module, which generates a 256-dimensional embedding. The Audio Transformer component of the model consists of four layers, with each layer having a hidden dimension of 256 and four attention heads for both self-attention and cross-attention. The MLP within the Audio Transformer layers utilizes Conv1d. The Post-Conv module uses Conv1d and Time-Aware Location-Variable Convolution with 32 channels.

Table 1: Comparison with other convential nerual vocoders in terms of quality and synthesis speed with the model trained on single-speaker dataset, LJSpeech.

| Model | Quality | | | | Speed |
| | MOS ($\uparrow$) | MCD ($\downarrow$) | V/UV ($\downarrow$) | F0 CORR ($\uparrow$) | RTF ($\downarrow$) |
| --- | --- | --- | --- | --- | --- |
| GT | 4.47$\pm$0.07 | / | / | / | / |
| WaveNet (MOL) | **4.23$\pm$0.06** | **1.74** | **6.87%** | **0.89** | 91.27 |
| WaveGlow | 3.79$\pm$0.09 | 2.83 | 17.18% | 0.68 | 0.049 |
| HIFI-GAN V1 | 3.94$\pm$0.08 | 2.08 | 8.98% | 0.77 | **0.003** |
| WaveGrad (noise schedule) | 3.88$\pm$0.07 | 2.76 | 10.31% | 0.69 | 0.051 |
| FastDiff (4 steps) | 4.05$\pm$0.07 | 2.56 | 8.59% | 0.79 | 0.025 |
| LinDiff (1 steps) | 3.99$\pm$0.06 | 2.17 | 9.12% | 0.74 | 0.004 |
| LinDiff (3 steps) | 4.12$\pm$0.07 | 1.96 | 8.75% | 0.79 | 0.013 |
| LinDiff (100 steps) | 4.18$\pm$0.05 | 1.92 | 7.78% | 0.82 | 0.520 |

**Training and Evaluation**   For this particular experiment, we trained the LinDiff model until it reached 200k steps using the Adam optimizer (Kingma & Ba, 2017) with $\beta_1 = 0.9, \beta_2 = 0.98, \epsilon = 10^{-9}$. Both models were trained on 4 NVIDIA GeForce RTX 3090 GPUs, using randomly sampled audio clips that matched the maximum transformer length (we set it 3600, which means max audio length is $3600 * 64/22050 = 10.44$ s), with a total batch size of 16. Initially, we trained the model for 10k steps without adversarial training. Every 5 steps, we update the weights of the discriminator and add a weight of 0.2 to LinDiff's adversarial loss to speed up the training process.

For the subjective evaluation of our system, we employ mean opinion scores (MOS) to assess the naturalness of the generated speech. The MOS were rated on a 1-to-5 scale and we report them along with the 95% confidence intervals (CI). In addition, we perform objective evaluations using several metrics, including Mel-cepstrum distortion (Kubichek, 1993) (MCD), error rate of voicing/unvoicing flags (V/UV), and correlation factor of F0 (F0 CORR) between the synthesized speech and the ground truth. To explore the diversity between the generated and real speeches, we calculate the Number of Statistically-Different Bins (NDB) and JensenShannon divergence (JSD). Furthermore, we evaluate the inference speed of our system on a single NVIDIA GeForce RTX 3090 GPU using the real-time factor (RTF).

## 4.2   COMPARISON WITH OTHER MODELS

We compared the proposed model in audio quality, diversity and sampling speed with other speech synthesis model, including 1) WaveNet(Oord et al., 2016), an autoregressive generative model. 2) WaveGlow(Prenger et al., 2019), a flow-based model. 3) HIFI-GAN V1(Kong et al., 2020a), a GAN-based model. 4) WaveGrad(Chen et al., 2020) and FastDiff(Huang et al., 2022), recently proposed DDPMs-based model.

The audio quality and sampling speed results are presented in Table 1. Our proposed method demonstrates the ability to synthesize high-fidelity speech with a limited number of steps. Even with just 3 steps, our model can generate speech of comparable quality to that produced by the autoregressive model, WaveNet (Oord et al., 2016). However, the inference speed surpasses that of WaveNet and other conventional vocoders. In fact, its speed is on par with HIFI-GAN while outperforming it in terms of the quality of generated samples. Regarding sample diversity, Table 2 shows that although LinDiff still lags behind the autoregressive model, WaveNet, it achieves greater variety in the generated speeches compared to other conventional vocoders.

## 4.3   ZERO-SHOT EXPERIMENT

To further investigate the capabilities of our model, we trained it on the multi-speaker dataset, LibriTTS. We evaluated it using the Mel-spectrograms extracted from LJSpeech. Since the speakers in LJSpeech were not used during the training of this experiment, this task is referred to as Zero-shot speech generation. The results are presented in table 3 below. It is evident from the table that all

Table 2: Comparison with other convential nerual vocoders in terms of diversity with the model trained on single-speaker dataset, LJSpeech.

| Model | NDB ($\downarrow$) | JSD ($\downarrow$) |
|---|---|---|
| GT | / | / |
| WaveNet (MOL) | **42** | **0.002** |
| WaveGlow | 115 | 0.011 |
| HIFI-GAN V1 | 69 | 0.004 |
| WaveGrad (noise schedule) | 122 | 0.007 |
| FastDiff (4 steps) | 65 | 0.005 |
| LinDiff (1 steps) | 81 | 0.005 |
| LinDiff (3 steps) | 71 | 0.004 |
| LinDiff (100 steps) | 58 | 0.004 |

Table 3: Comparison with other conventional nerual vocoders in terms of speech quality with the model trained on multi-speaker dataset. Zero-Shot MOS means we test the model on speakers that is not included in the training set.

| Model | MOS ($\uparrow$,Seen speaker) | Zero-Shot MOS ($\uparrow$) |
|---|---|---|
| GT | 4.47±0.07 | 4.47±0.07 |
| WaveNet (MOL) | **4.10±0.08** | **4.03±0.07** |
| WaveGlow | 3.64±0.06 | 3.56±0.07 |
| HIFI-GAN V1 | 3.94±0.08 | 3.89±0.06 |
| WaveGrad (noise schedule) | 3.74±0.07 | 3.71±0.06 |
| FastDiff (4 steps) | 3.95±0.07 | 3.90±0.08 |
| LinDiff (1 steps) | 3.73±0.08 | 3.68±0.07 |
| LinDiff (3 steps) | 3.83±0.08 | 3.71±0.08 |
| LinDiff (100 steps) | 3.92±0.07 | 3.85±0.06 |

models perform worse when tasked with generating speech from multiple speakers. Our proposed transformer-based model suffers a significant performance drop in this scenario. The reason may be, in the case of a single-speaker dataset, the token embedding space is smaller than that of a multi-speaker dataset, as we split the input audio into small patch tokens. Therefore, modeling the latter is more challenging.

### 4.4 ABLATION STUDY

We conducted separate experiments to evaluate the performance of our model after removing the Post conv layer or discarding adversarial training. The results shows performance decrease. This is particularly evident in the spectrogram analysis of synthesized speech, which exhibits harmonic noise patterns as distinct horizontal lines when either component is removed. In addition, we explore different sampling steps' influence on the final results. We compare the synthesized audio with 1 step and 100steps. Fig 2 shows the results. We also investigate the impact of patch size on our model. The results, as presented in Table 4, confirm that smaller patch sizes indeed yield higher quality samples. However, considering sampling efficiency, it is advisable to choose a sufficiently large patch size. After careful consideration, we select a patch size of 64 as it strikes the best balance between sample quality and sampling speed.

## 5 DIFFERENCE WITH GAN

Actually, the one step diffusion model (we mean during training using one step) is just a GAN model, we name it LinDiff (GAN). We compared it with the model trained with 1000steps. both the quality and diversity of the generated samples are evaluated. Results can be found in table 5. The

Table 4: Ablation study results. Comparison of models with different configs. We set sampling steps to 100 in this experiment.

| Model | MOS (↑) | MCD (↓) | V/UV (↓) | F0 CORR (↑) |
|---|---|---|---|---|
| LinDiff origin (patch 64) | **4.18±0.05** | **1.92** | **7.78%** | **0.82** |
| LinDiff w/o Post-Conv | 3.84±0.06 | 2.98 | 9.36% | 0.72 |
| LinDiff w/o adv training | 3.74±0.08 | 2.91 | 10.39% | 0.76 |
| LinDiff (patch 128) | 3.65±0.08 | 2.87 | 16.57% | 0.67 |
| LinDiff (patch 256) | 3.34±0.05 | 3.34 | 17.89% | 0.59 |

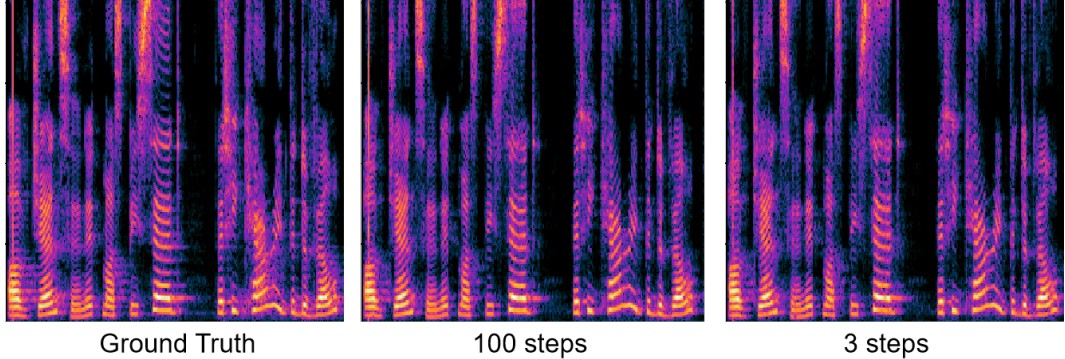

| Ground Truth | 100 steps | 3 steps |

Figure 2: Visualization of spectrograms from the ground truth audio and predicted audio with 100 steps and 3 steps.

sample generated from LinDiff (GAN) has limited diversity (the Statistically-Different Bins (NDB) and JensenShannon divergence (JSD) reflects). This proves the combination improves the diversity significantly.

## 6 CONCLUSION

In this work, we present LinDiff, a novel conditional diffusion model designed for fast and high-fidelity speech synthesis. By leveraging an Ordinary Differential Equation (ODE), we construct a diffusion path that offers improved fitting capabilities with reduced sampling steps compared to previous DDPMs. Moreover, LinDiff incorporates Transformer and CNN architectures. The Transformer model captures global information at a coarse-grained level, while the Convolutional layers handle fine-grained details. Additionally, we employ generative adversarial training to further enhance sampling speed and improve the quality of synthesized speech. Experimental results demonstrate that the proposed method can synthesis speech of comparable quality to autoregressive models, with a Real-Time Factor (RTF) of 0.013, making it significantly faster than real-time usage. (Le et al., 2023)

Table 5: Comparison between LinDiff (GAN) with LinDiff.

| Model | MOS (↑) | NDB (↓) | JSD (↓) |
|---|---|---|---|
| LinDiff (GAN) | 4.01±0.05 | 99 | 0.007 |
| LinDiff (1 steps, trained with 1000 steps) | 3.99±0.06 | 81 | 0.005 |
| LinDiff (3 steps, trained with 1000 steps) | 4.12±0.07 | 71 | 0.004 |
| LinDiff (100 steps, trained with 1000 steps) | **4.18±0.05** | **58** | **0.004** |

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
