# OpenReview forum: "Boosting Fast and High-Quality Speech Synthesis with Linear Diffusion"
_ICLR.cc/2024/Conference — Submitted to ICLR 2024_

### Official Review · Reviewer_QZgE · 2023-10-30

**Soundness:** 2 fair
**Presentation:** 2 fair
**Contribution:** 2 fair
**Rating:** 3
**Confidence:** 4

**Summary:**

This paper proposes a linear diffusion model (LinDiff) to synthesize waveform from mel-spectrogram, aiming to achieve fast inference speed and high sample quality. This paper contains two contributions, the first is to model the waveform based on Rectified-Flow, the second is to divide the waveform into patches, and design a model structure similar to ViT. In terms of the subjective indicator (MOS), the Vocoder proposed in this paper slightly exceeds HifiGAN and other Diffusion-based Vocoder (such as WaveGrad and FastDiff).

**Strengths:**

This paper contains two strengths,

First of all, the application of the structure of ViT in the waveform field is relatively novel, and this attempt should be encouraged. In theory, the model incorporates the use of a patch-wise token and the Transformer architecture for effective modeling of global information in noisy speech. This helps in capturing the contextual dependencies and improves the overall synthesis quality.

Secondly, from Table.1 in the experimental part, we can simply think that the newly proposed Vocoder has reached a new SOTA in terms of MOS.

**Weaknesses:**

## Weakness 1
The idea of "linear diffusion" in this paper basically comes from Rectified-Flow. The authors just apply Rectified-Flow to the audio field, and there is nothing new in machine learning theory. However, it is a pity that some works have already applied Flow-matching technology into the audio field, such as [1].

[1] Voicebox: Text-Guided Multilingual Universal Speech Generation at Scale

## Weakness 2
The experimental part of the paper is not convincing.

  For eg.

  HIFI-GAN V1 3.94±0.08 (MOS)

  LinDiff (1 steps) 3.99±0.06 (MOS)

  It is difficult to say that it has an advantage over the HiFiGAN model (LinDiff only gains 0.05±0.08).

HIFI-GAN V1     2.08 (MCD↓)

LinDiff (1 steps) 2.17 (MCD↓)

LinDiff is worse than HiFiGAN in terms of objective indicators.

## Weakness 3
The authors mentioned that this paper uses Transformer structure to model waveform for the first time, and the advantage of this structure is "capturing the contextual dependencies", so why is there no relevant experiment to prove the superiority of Vocoder over other Vocoders in contextual modeling?

## Weakness 4
In the current research environment, LjSpeech, a small lightweight dataset, is no longer enough to verify the superiority of the model (because everyone's scores are very high). Table 3 reveals that LinDiff performs poorly on large datasets such as Libritts.

**Questions:**

None

---

> ### Author Response · Authors · 2023-11-11
> **Rebuttal**
>
> We thank the reviewer for recognizing the positive aspects of our paper, and we will address the reviewer’s concerns in the following parts.
>
> **Q1:** About linear diffusion.
>
> **R1:** In Rectified-Flow [1], the authors didn't prove that the diffusion path built with linear interpolation is a Normalizing Flow. We combine both the Rectified-Flow and Flow-matching [2] to prove this. Also Voicebox: Text-Guided Multilingual Universal Speech Generation at Scale is a work done three months ago while its paper is released on 19 Oct. Our work is done earlier.
>
> **Q2:** About the experimental part.
>
> **R2:** As you can see in the previous work like FastDiff [3], WaveGrad [4]. The diffusion based model struggles to decrease the sampling steps needed. To our knowledge, FastDiff takes 4 steps to reach good quality. Actually, as you can see our model takes only 3 steps to ahcieve corresponding good results. We here show the 1-step result to explain that our model can even generate greate result with only 1 step, beating most of the method using diffusion though it meets small performance degradation.
>
> **Q3:** About the Transformer+post conv structure
>
> **R3:** We provide the comparision between our proposed structure and that in FastDiff. Here we remove the adversial training to explore the structure's influence.
> |        Model        |    MOS   |     MCD ($\downarrow$)  |       V/UV ($\downarrow$)  |       F0 CORR($\uparrow$) |
> |:-|:-|:-|:-|:-|
> | LinDiff(1000 steps,w/o adv training, FastDiff structure)  | 4.01±0.06   |     2.23    |     8.68%    |     0.79    |
> |  LinDiff (1000 steps, w/o adv training, Transformer structure) | 4.04±0.07 | 2.09   |      8.63%     |     0.79     |
> | LinDiff(100 steps,w/o adv training, FastDiff structure)  |3.72±0.06   |     2.97    |     10.53%    |     0.76    |
> |  LinDiff (100 steps, w/o adv training, Transformer structure) |  3.74±0.08 | 2.91   |      10.39%     |     0.76     |
> | LinDiff(1 step,w/o adv training, FastDiff structure)  |  3.66±0.06   |     3.02    |     11.95%    |     0.73    |
> |  LinDiff (1 step, w/o adv training, Transformer structure)| 3.65±0.05 | 3.03   |      11.63%     |     0.73     |
>
> From above results, we can find that they reach similar speech quality. However, with 4 steps transformer-structure's real-time factor is about 0.017 (FastDiff gives the 4 steps for inference). While the conv structure （FastDiff structure） is about 0.025. This means our structure ahieves about 1.56 times's faster inference speed.
>
> **Q4:** Results on large sacle dataset.
>
> **R4** As shown in the paper, we use a hidden size of 256 in the experiment. This is enough for Ljspeech, however for more complex dataset like Libritts, it is too small. After we increase the hidden size to 384, the quality improves a lot. We are sorry for not to fit the best parameter in our paper and thus raise some misunderstanding.
>
> Thanks agagin, we hope this can address your concern

---

### Official Review · Reviewer_zJkF · 2023-10-31

**Soundness:** 3 good
**Presentation:** 3 good
**Contribution:** 3 good
**Rating:** 5
**Confidence:** 4

**Summary:**

The paper proposes a linear diffusion model (LinDiff) based on an ODE to simultaneously reach fast inference and high sample quality. The two main components of the Lindiff is an ODE formulation to enable linear interpolation and a Transformer based model on ground-truth (wav_gt) prediction. Experiments on LJSpeech and LibriTTS show the effectiveness of proposed method over previous baselines.

**Strengths:**

1. The paper propose an ordinary differential equation formulation on waveform generation, which can help model to generate relatively high-fidelity speech with limited steps.

2. The paper firstly introduce a Transformer based noise predictor for waveform generation.

3. Experiments and ablation study show that the Lindiff is better than the previous baselines.

**Weaknesses:**

The paper is well-written and clear. I acknowledge the contributions of the paper on ODE formulation and Transformer-based noise predictor. However, if these are the main contributions, I think more experiments should be conducted to verify the effectiveness of proposed method.

1. As for the ODE formulation, apart from the proposed formulation, there exists many other formulation (e.g., ODE in Grad-TTS/NaturalSpeech 2 and the original DDPM), which can also predict the ground-truth waveform. I think ablation on formulation (while keep the GAN and Transformer predictor be the same) is necessary to verify the contribution of proposed formulation.

2. As for the noise predictor, it is necessary to compare Transformer-based predictor with the convolutional based predictor (e.g., WaveNet based or Unet based) while keep the GAN and ODE formulation be the same to verify the effectiveness of Transformer-based predictor.

**Questions:**

1. The paper should give a more detailed description about patch (e.g., how to transform waveform to patch and how to transform predict path to waveform). According to the size in Section 3.4, it seems that the waveform is transformed to 256-dim STFT before formulating batch?

2. Since patch seems to be a very sensitive parameter, it will be better if the ablation study on patch can be more detailed to show the trade-off (adding experiments of patch=16 and 32).

---

> ### Author Response · Authors · 2023-11-11
> **Rebuttal**
>
> We sincerely appreciate your hard work and positive comments on our paper. We will address your concerns in the following parts.
>
> **Q1:** About the patch.
>
> **R1:** We turn the 64 continuous wav points (non-overlap) as a patch and use a linear projection to turn them into 256-dim input. When we fit the model to predict the diffusion path, we actually enable the model to predict 0.001*(target - noise) (this is the conditional flow-matching objective used for training), we could iteratively reconstruct the original wav by repeatly add the predicted 0.001*(target - noise) to the noise audio and predict next 0.001*(target - noise)  till time T (1000, or if we want to sample with 100 steps then every time, we add 10*the predicted 0.001*(target - noise)  to our noise audio).
>
> **Q2:** About the structure.
>
> **R2:** We provide the comparision between our proposed structure and that  with convolutional based predictor (As FastDiff use a conv structure, we directly use their structure). Here we remove the adversial training to explore the structure's influence.
> |        Model        |    MOS   |     MCD ($\downarrow$)  |       V/UV ($\downarrow$)  |       F0 CORR($\uparrow$) |
> |:-|:-|:-|:-|:-|
> | LinDiff(1000 steps,w/o adv training, FastDiff structure)  | 4.01±0.06   |     2.23    |     8.68%    |     0.79    |
> |  LinDiff (1000 steps, w/o adv training, Transformer structure) | 4.04±0.07 | 2.09   |      8.63%     |     0.79     |
> | LinDiff(100 steps,w/o adv training, FastDiff structure)  |3.72±0.06   |     2.97    |     10.53%    |     0.76    |
> |  LinDiff (100 steps, w/o adv training, Transformer structure) |  3.74±0.08 | 2.91   |      10.39%     |     0.76     |
> | LinDiff(1 step,w/o adv training, FastDiff structure)  |  3.66±0.06   |     3.02    |     11.95%    |     0.73    |
> |  LinDiff (1 step, w/o adv training, Transformer structure)| 3.65±0.05 | 3.03   |      11.63%     |     0.73     |
>
> From above results, we can find that they reach similar speech quality. However, with 4 steps transformer-structure's real-time factor is about 0.017 (FastDiff gives the 4 steps for inference). While the conv structure （FastDiff structure） is about 0.025. This means our structure ahieves about 1.56 times's faster inference speed.
>
> **Q3:** About the patch size.
>
> **R3:** With a patch size of 64, our model can generates speech with fast speed, however, if we further decrease the patch size, the compution cost may be unbearable (as the audio_length=mel-spectrem*256, we here use th patch size that is factor of 256, so if choose patch size smaller than 64, it will be 32). Though it makes good result, the real-time factor becomes about 5 times bigger (5 times time consumption.)

---

> > ### Comment · Reviewer_zJkF · 2023-11-22
> > **Thanks for your response**
> >
> > Thanks for your effort in rebuttal. I decide to keep my score due to the following reasons:
> >
> > 1. The ablation study in model architecture shows that the improvement of new architecture is quite marginal, especially when the step is small (which is the application scenario of the proposed method).
> >
> > 2. I still think the ablation on ODE formulation is necessary (Weakness 1 in response).
> >
> > By the way, it is acceptable that smaller patch size can improve the quality at the cost of latency. So it will be better if the author can report this in detail in the next version of the paper rather than a description.

---

### Official Review · Reviewer_2DZz · 2023-11-01

**Soundness:** 3 good
**Presentation:** 3 good
**Contribution:** 3 good
**Rating:** 6
**Confidence:** 4

**Summary:**

This paper proposes a linear diffusion model to reach fast inference speech and high sample quality. The authors demonstrate that its synthesis quality is on par with autoregressive vocoders while offering faster synthesis speed. They also introduce a patch-based processing approach to reduce computational complexity.

**Strengths:**

As far as I checked, the proposed LinDiff is technically sound. The proposed network architecture is novel. The experimental results suggest that LinDiff is capable of generating high-quality speech even with one sampling step. In the demo page, from a subjective feeling, the quality of LinDiff is better than FastDiff.

**Weaknesses:**

**Presentation**: It is quite hard to proceed from the section 2 (background) to the section 3 (method). I believe there are some irrelevant formulas (e.g. Eq. (4)) in section 2 that does not contribute to the design of LinDiff. These formulas might sidetrack and, to a large extent, hinder readers' understanding. A quick fix would be to cite the contents from another paper and only keep the most influential ones (e.g. Eq. (8)). Besides, I cannot find the training loss for stage 1 in Algorithm 1, please correct it for a self-contained presentation. Also, I suggest to bold all vectors and matrices, following the usual practice of ICLR papers, to differentiate them from the scalars.

**More ablations**: I am also skeptical of the contribution of different novel points to the final performance of LinDiff, including the new architecture, three-stage training and the use of adversarial loss. Especially concerning the new architecture, I recommend the authors to compare it with some widely used architecture, e.g., UNet1D, or DiffWave. I believe objective measures such as MCD would be sufficient to confirm the superiority of the proposed architecture.

**Questions:**

My questions are stated above.

---

> ### Author Response · Authors · 2023-11-11
> **Rebuttal**
>
> We thank the reviewer for recognizing the contributions of our paper and giving us constructive suggestions. We will address the reviewer’s concerns in the following parts.
>
> **Q1:** About the Presentation.
>
> **R1:** We apologize for the confusion. We want to briefly introduce the theory about normalizing flow. We will rewrite this parts to make it more acceptable. We are sorry for missing the stage 1 loss. It is just $MSE(a_T^{rev}-a_T)+MSE(STFT(a_T^{rev})-STFT(a_T))$. We will add this to our paper. Also we thank you for providing us with such valuable suggestion.
>
> **Q2:** About the structure.
>
> **R2:** We provide the comparision between our proposed structure and that in FastDiff (this is a conv based model, and achieves SOTA results before our model). Here we remove the adversial training to explore the structure's influence.
> |        Model        |    MOS   |     MCD ($\downarrow$)  |       V/UV ($\downarrow$)  |       F0 CORR($\uparrow$) |
> |:-|:-|:-|:-|:-|
> | LinDiff(1000 steps,w/o adv training, FastDiff structure)  | 4.01±0.06   |     2.23    |     8.68%    |     0.79    |
> |  LinDiff (1000 steps, w/o adv training, Transformer structure) | 4.04±0.07 | 2.09   |      8.63%     |     0.79     |
> | LinDiff(100 steps,w/o adv training, FastDiff structure)  |3.72±0.06   |     2.97    |     10.53%    |     0.76    |
> |  LinDiff (100 steps, w/o adv training, Transformer structure) |  3.74±0.08 | 2.91   |      10.39%     |     0.76     |
> | LinDiff(1 step,w/o adv training, FastDiff structure)  |  3.66±0.06   |     3.02    |     11.95%    |     0.73    |
> |  LinDiff (1 step, w/o adv training, Transformer structure)| 3.65±0.05 | 3.03   |      11.63%     |     0.73     |
>
> From above results, we can find that they reach similar speech quality. However, with 4 steps transformer-structure's real-time factor is about 0.017 (FastDiff gives the 4 steps for inference). While the conv structure （FastDiff structure） is about 0.025. This means our structure ahieves about 1.56 times's faster inference speed.

---

### Official Review · Reviewer_JAcM · 2023-11-02

**Soundness:** 2 fair
**Presentation:** 3 good
**Contribution:** 2 fair
**Rating:** 3
**Confidence:** 5

**Summary:**

This paper proposes a linear diffusion model (LinDiff), a fast and high-fidelity speech synthesis based on conditional diffusion models with an ordinary differential equation. LinDiff incorporates Transformer and CNN architectures for effective modeling of global information and refining details. LinDiff can synthesize high-quality speech conditioned on mel-spectrograms with only one diffusion step.

**Strengths:**

This model uses a linear diffusion process with a flow matching training method to model speech synthesis. Experiments show that it can generate higher-quality results with fewer denoising steps. The proposed model can synthesize speech with quality comparable to the autoregressive models with faster speed.

**Weaknesses:**

1. The main weakness of this paper is the lack of innovation. The key point of the paper is using a linear diffusion process with flow matching; however, this has been proposed in previous work and shown to significantly reduce the number of inference steps.

2. The authors did not prove the impact of the state incorporating Transformers and CNN architectures on the results. For example, using Transformers as the backbone of diffusion is not necessarily necessary, and authors should compare it with CNN-based architectures. In fact, I don’t think using a framework like VIT is necessary for the task of vocoder. Adding self-attention to CNN architectures (e.g. WaveNet) may have similar results.

3. The experimental results did not show obvious improvement. In the case of single-step diffusion, MCD, V/UV, and F0 CORR are not as good as HIFIGAN. Note that HIFIGAN is no longer a strong baseline in the vocoder field.

4. Comparisons of objective measures of diversity deserve further discussion. I think for the task of vocoder, diversity is not an important evaluation criterion, and it may not make any difference to people's sense of hearing.

**Questions:**

1. The author uses discrete time steps in the process of training the model. I would like to ask the author whether he has tried using sampling continuous time.

2. Can the author provide more detailed experiments to verify the necessity of Transformers as the backbone of diffusion modeling?

3. I think this framework looks like it should serve as a general conditional speech synthesis model. However, the author only conducted experiments on the vocoder task. Can the authors verify the feasibility of the framework on more tasks (e.g., TTS)?

---

> ### Author Response · Authors · 2023-11-11
> **Rebuttal**
>
> We thank all of the reviewers' efforts and will address some of the issues proposed by reviewers here.
>
> **Q1:** About sampling continuous time.
>
> **R1:** I am sorry but we have not tried a continuous manner. When we reconstruct, we use the euler method to approximate the integration process in the ODE as it can not be directly computed. If we use a continuous manner, then the time t becomes a continuous variable, we can not build a time embedding for it. Besides we still needs the Euler to solve the ODE problem. Because of the problem mentioned before, we didn't take continuous time.
>
> **Q2:** About the transformer structure.
>
> **R2:** We provide the comparision between our proposed structure and that in FastDiff (This is the previous SOTA diffusion-based model, it utilizes a Conv structure.). Here we remove the adversial training to explore the structure's influence.
> |        Model        |    MOS   |     MCD ($\downarrow$)  |       V/UV ($\downarrow$)  |       F0 CORR($\uparrow$) |
> |:-|:-|:-|:-|:-|
> | LinDiff(1000 steps,w/o adv training, FastDiff structure)  | 4.01±0.06   |     2.23    |     8.68%    |     0.79    |
> |  LinDiff (1000 steps, w/o adv training, Transformer structure) | 4.04±0.07 | 2.09   |      8.63%     |     0.79     |
> | LinDiff(100 steps,w/o adv training, FastDiff structure)  |3.72±0.06   |     2.97    |     10.53%    |     0.76    |
> |  LinDiff (100 steps, w/o adv training, Transformer structure) |  3.74±0.08 | 2.91   |      10.39%     |     0.76     |
> | LinDiff(1 step,w/o adv training, FastDiff structure)  |  3.66±0.06   |     3.02    |     11.95%    |     0.73    |
> |  LinDiff (1 step, w/o adv training, Transformer structure)| 3.65±0.05 | 3.03   |      11.63%     |     0.73     |
>
> From above results, we can find that they reach similar speech quality. However, with 4 steps transformer-structure's real-time factor is about 0.017 (FastDiff gives the 4 steps for inference). While the conv structure （FastDiff structure） is about 0.025. This means our structure ahieves about 1.56 times's faster inference speed.
>
> **Q3:** About the one step model.
>
> **R3:** As you can see in the previous work like FastDiff [3], WaveGrad [4]. The diffusion based model struggles to decrease the sampling steps needed. To our knowledge, FastDiff takes 4 steps to reach good quality. Actually, as you can see our model takes only 3 steps to ahcieve corresponding good results. We here show the 1-step result to explain that our model can even generate greate result with only 1 step, beating most of the method using diffusion though it meets small performance degradation.
>
> **Q4:** About apply it for TTS.
>
> **R4:** Thanks for your suggestion. Actually, there are some works actually done this "VoiceFlow: Efficient Text-to-Speech with Rectified Flow Matching"[1].
> 1. Guo, Yiwei, et al. "VoiceFlow: Efficient Text-to-Speech with Rectified Flow Matching." arXiv preprint arXiv:2309.05027 (2023).
> We hope to verify this method as vocoder as many previous like FastDiff, wavegrad all verify their work with mel to audio.

---

### Meta-Review · Area_Chair_m3fv · 2023-12-05

**Metareview:**

The authors introduce a linear diffusion model for fast and high-quality speech synthesis.  This ODE based model can efficiently reduce the number of steps required during inference.  The authors also proposed to use a transformer architecture to capture long context information.  Adversarial training is also used to further improve the inference speed and sample quality.  The work is well motivated and may have its value to the community.  The authors also cleared up some of the concerns raised by the reviewers in their rebuttal.  Although reviewers agree that there is some novelty in the work, they also consider the novelty not overwhelmingly significant given existing related work.  Even with the added ablation study in the rebuttal, the improvement under the proposed techniques seems to be incremental.  Given its current form, this paper does not meet the acceptance threshold.

**Justification For Why Not Higher Score:**

The novelty is not overwhelmingly significant.  The performance improvement appears to be incremental.

**Justification For Why Not Lower Score:**

N/A

---

### Decision · Program_Chairs · 2024-01-16

Reject